# What Are the Factors That Enable Thread Lifting to Last Longer?

**Gi-Woong Hong** [1,†]**, Hyewon Hu** [2,†]**, Soo-Yeon Park** [3]**, Jovian Wan** [4] **and Kyu-Ho Yi** [2,5,]*

1   Sam Skin Plastic Surgery Clinic, Seoul 06577, Republic of Korea; cosmetic21@hanmail.net
2   Division in Anatomy and Developmental Biology, Department of Oral Biology, Human Identification
    Research Institute, BK21 FOUR Project, Yonsei University College of Dentistry, 50-1 Yonsei-ro, Seodaemun-gu,
    Seoul 03722, Republic of Korea; wonhuh@yuhs.ac
3   Made-Young Plastic Surgery Clinic, Seoul 06615, Republic of Korea; sy-jumsim@hanmail.net
4   Asia Pacific Aesthetic Academy, Hong Kong; jovian.wan@apaa.org
5   Maylin Clinic (Apgujeong), Seoul 06001, Republic of Korea
*   Correspondence: kyuho90@daum.net; Tel.: +82-2-2228-3047; Fax: +82-2-393-8076
†   These two authors contributed equally as first authors.

**Abstract:** Thread-lifting traditionally addressed aging-related skin laxity by leveraging precise thread placement and traction. However, recent advancements, notably cog threads, expanded its application to younger patients seeking facial contour refinement. These newer threads effectively lift sagging areas and refine facial contours, broadening the procedure's appeal. Challenges arise in selecting threads due to variable physician preferences and patient needs. Clear indications for thread efficacy are vital for credibility and tailored selection. Thread choice depends on tissue laxity, necessitating lighter threads for minimal laxity and stronger ones for significant sagging. However, no single thread universally suits all cases. Combining different threads is favored for optimal outcomes and minimizing side effects. Excessive traction post-procedure may lead to prolonged discomfort and skin irregularities. Post-procedural tension adjustments through massage remain debated, potentially conflicting with minimally invasive principles. Understanding thread characteristics guides tailored selection, considering patient conditions and procedural goals. This comprehensive understanding extends beyond specific products, aiming for optimal outcomes in thread-lifting procedures. Key factors influencing outcomes encompass thread materials, thickness, cog shapes, insertion depth, lifting vectors, and absorbable thread expiration dates.

**Keywords:** thread lifting; cog threads; facial contour; thread selection; minimally invasive; thread characteristics

## 1. Introduction

Until recently, the primary application of thread lifting was chiefly aimed at addressing skin and connective tissue laxity that occurs concomitantly with the aging process. Anatomically, the focus was on comprehending why and how such laxity occurs in these tissues as aging progresses, particularly the structural alterations within the facial anatomy that are significantly involved in these changes [1]. Consequently, in thread-lifting procedures, the primary concern revolved around efficiently ameliorating the sagging of skin and connective tissue by employing specific threads inserted into particular planes, exerting traction on specific tissues to achieve a smoother appearance and contribute to a more youthful facial aesthetic [2–4].

However, techniques utilizing cog threads have evolved in recent times not only for aging patients seeking improvement in the midface area and perioral region to address jowls and refine the jawline but also for relatively younger patients desiring a more slender and taut facial contour, despite not presenting significant skin laxity [5]. Moreover, the latest iterations of threads not only exhibit effects in lifting sagging cheeks and perioral areas but also display efficacy in creating a slender facial appearance, consequently broadening the patient demographic. Notably, this expansion includes not just individuals within an age

bracket manifesting a square-shaped face due to reduced skin elasticity, but also younger patients in their twenties and thirties aspiring for a smaller and more streamlined facial appearance [6–8]. For younger patients with facial features that, while not inherently square or large, still convey a broad facial impression, targeted lifting in the areas of widened or sagging cheeks and jawlines provides an observable effect in sculpting a smaller and more refined facial contour, an effect that may not be achievable solely through square jaw botulinum neurotoxin injections [9–11].

It is challenging to establish precise criteria for selecting threads based on the extent of skin and tissue laxity when considering thread-lifting procedures. The difficulty arises from the variability in the physician's pursuit of the desired level of traction, compounded by individual patient preferences. In practice, certain thread manufacturers promote dramatic effects by showcasing before-and-after images featuring cases suitable for the procedure [12,13]. However, for enhanced credibility, providing accurate indications regarding when the product may be beneficial and when its efficacy might be limited in specific cases could assist in selecting the appropriate threads and procedural methods tailored to individual patient cases [5,14–17]

Once certain personal benchmarks are established, the choice of threads depends on the degree of tissue laxity. For instances where minimal traction force is required due to lesser laxity, lighter cog threads that cause minimal discomfort or foreign body sensation could be suitable. Conversely, when dealing with considerable laxity requiring stronger tensile strength and pulling force, selecting products that match these requirements becomes imperative. Choosing longer U-shaped or V-shaped threads with superior pulling force or I-shaped threads with robust traction and maintaining force (considering thread thickness, cog traction, sustaining capabilities, etc.) becomes necessary, especially if methods involving tying or securing to a mesh are not utilized. However, it is important to note that no single thread type universally suits all cases. For instance, while U-shaped threads may offer effective traction in the temporal region, patient discomfort levels can vary. Additionally, compared to shorter I-shaped threads, the penetration and exit of the thread ends through the skin during the procedure may lead to bleeding, skin dimpling, or irregularities, necessitating careful consideration when selecting the product [4,18].

The ideal approach aims to minimize patient discomfort and side effects while achieving an optimal outcome with appropriate products. To fulfill this objective, current trends suggest a move away from solely utilizing a single type of thread towards combining various threads with differing mechanisms of action, as previously explained. Excessive traction during procedures can lead to prolonged discomfort lasting over 1–2 months, and the unevenness of the skin may persist even longer. To mitigate such side effects, some practitioners suggest initially applying maximum traction and subsequently employing massage to loosen the tissues, thereby purportedly allowing for adjustment of the applied force. However, the efficacy of this approach remains variable. Excessive post-procedural massage might inadvertently release the intentionally applied tension, resulting in a loss of the intended effect. Conversely, in certain cases, despite extensive massage, tissue release might not occur, potentially eliciting significant patient complaints. Consequently, this approach may not align with the concept of minimally invasive procedures aimed at easily manipulating tissues without resorting to traditional face-lifting surgeries [19,20].

Therefore, depending on the types and forms of threads utilized in thread-lifting procedures, each thread exhibits its own set of advantages and disadvantages. Consequently, it is imperative for us to thoroughly understand the characteristics of each type of thread and the quality of thread products offered by different companies. This understanding allows for the selection of threads that align with the patient's condition, the objectives of the procedure, and the conceptual approach pursued by the practitioner. To this end, the author aims to examine various factors influencing the outcomes of thread-lifting procedures, which are crucial to comprehend irrespective of the specific thread products offered by different companies, in order to achieve optimal results in thread-lifting procedures.

## 2. Patient Selection

Achieving optimal and long-lasting results in thread lifting is closely connected to the careful selection of patients and the customization of the procedure based on each individual's unique degree and direction of sagging [21]. This personalized approach necessitates effective communication between physicians and patients during the design phase to ensure mutual understanding and satisfaction. Identifying key vectors, including the oblique, vertical, and nasolabial vectors, significantly influences the direction and degree of skin lifting. Emphasizing the simulation of the lifting process during consultation, using multiple fingers or the palm, is vital for accurate outcome prediction, aiding physicians in comprehending potential results and communicating effectively with the patient. A thorough discussion of thread type and quantity during consultation is recommended to effectively manage patient expectations.

Yi et al. highlight the critical importance of a secure fixing point, particularly in the temporal fascia area, for achieving optimal lifting effects that result in enduring outcomes. Various thread-lifting techniques, each with its own method of creating a fixing point, are discussed in this article, underscoring the need for a method that ensures a strong fixing point with minimal difficulty and side effects. This emphasis contributes to the overall safety and effectiveness of the thread-lifting procedure [22].

## 3. Side Effects of Thread-Lifting

Irrespective of the material used for thread lifting, complications are not uncommon, as reported in the literature and our clinical experience [23,24]. These complications encompass a spectrum of issues, spanning short-term symptoms such as bruising, pain, swelling, bleeding, and hematoma formation, alongside aesthetic concerns including skin dimpling, irregularity, abnormal facial contour, and patient dissatisfaction. Neurosensory sequelae, such as tension, numbness, and pruritus, are reported, along with complications like infection, inflammation, abscess, thread protrusion and migration, subcutaneous induration, and granuloma. Documented injuries to surrounding structures include motor nerve paralysis, parotid gland, and duct [21,24–26]

While many complications are mild and may spontaneously resolve or be addressed through nonsurgical interventions, specific issues, such as thread protrusion, subcutaneous nodules, and infection, may necessitate surgical intervention involving the removal of threads and debridement [27]. The most frequently noted complications are persistent pain, thread protrusion, dimpling, sensory abnormalities, and foreign body reactions. A study by Rachel et al. [28]. highlighted that the risk of complications appears to increase with the number of threads employed [26]. While the chronic inflammatory reactions mentioned are infrequent, the literature underscores the potential for repetitive trauma and micro-movements between the barbs of the sutures and the surrounding capsule to induce chronic inflammation in the facial soft tissues [19]. All complications arising from thread lifting invariably extend the recovery period for patients and may potentially impact both aesthetic and functional outcomes [7,21].

## 4. The Materials of Threads

The choice of materials for threads used in thread lifting significantly influences the clinical effects and duration of maintenance during the lifting procedure. Initially, non-absorbable polypropylene (Prolene™) threads were extensively utilized in barbed suture lifting. However, in recent times, absorbable polydioxanone (PDO) threads have become the most prevalent. Additionally, there has been an emergence of absorbable threads made from materials such as poly-L-lactic acid (PLLA) and polycaprolactone (PCL). These absorbable threads are not solely produced from a single component but are also created by blending with substances like polyglycolic acid (PGA). For instance, the Quill lift thread is manufactured using a blend of PGA at 75% and PCL at 25%, while the V-loc thread is composed of PGA at 60%, polytrimethylene carbonate (PTMC, Maxon™) at 26%, and PDO at 14%.

Unlike non-absorbable threads primarily composed of a single component, such as polypropylene, the rationale for combining various materials in absorbable threads stems from the differing tissue responses, tensile strengths, and dissolution periods of each constituent material within the human body. For instance, threads like chromic catgut demonstrate favorable tissue response but possess relatively weak tensile strength and are quickly absorbed due to rapid dissolution. In contrast, PDO threads exhibit a robust tensile strength among absorbable threads and take over 6 months to dissolve, thus offering an extended duration of maintenance, contributing to their widespread usage [29,30].

The composition of threads also influences the elasticity exhibited by these threads. Here, elasticity refers to the elastic force, which signifies the force exerted by an elastic object to return to its original state. Elastic force is typically computed following Hooke's Law and is often represented by the formula $F = k\Delta X$ [31–33]. In this expression, F represents the elastic force, k signifies the elastic modulus, which quantifies the stiffness of the object, and $\Delta X$ denotes the extent of deformation, known as the elastic limit. It is important to note that elasticity should not be simplistically perceived as a property solely indicative of good resilience in an object; rather, it pertains to the force exerted by an object possessing elasticity to revert to its initial state.

If we delve into the equation, the elastic modulus (k) and elastic limit (X) are constants determined by the material and shape of an object. Consequently, when comparing objects that exhibit similar elastic forces (F), a material with a higher elastic modulus and a lower elastic limit feels more solid, akin to a rigid solid, whereas a material with a smaller elastic modulus but a larger elastic limit feels more flexible, commonly perceived as having good elasticity, resembling characteristics of a viscoelastic substance rather than a rigid material (Figure 1).

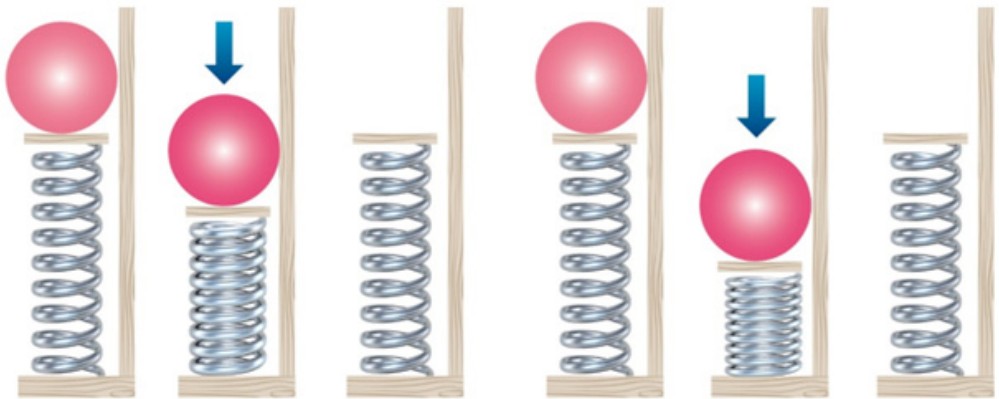

Large elastic modulus (K) & small elastic limit (X)     Small elastic modulus (K) & large elastic limit (X)

**Figure 1.** A comparison of two springs, each marked with arrows indicating the direction and magnitude of their elastic force (F). The left spring demonstrates higher stiffness and limited displacement, reflecting a higher elastic modulus (K) and lower elastic limit (X), resulting in a solid, rigid feel. The right spring, exhibits lower stiffness and greater displacement, representing a lower elastic modulus and higher elastic limit, showcasing flexibility and good elasticity, akin to viscoelastic properties.

For instance, Silhouette Soft threads, primarily composed of PLLA (poly-L-lactic acid), possess a relatively larger elastic limit compared to other threads [14,17,34,35] This characteristic allows them to adapt more effectively to the skin and tissue contraction and relaxation that occur when the face is in motion or during facial expressions. Consequently, discomfort arising from facial movements post-procedure might be mitigated to a lesser extent. However, it is important to note that threads exhibiting remarkable elasticity may not universally be deemed superior in all cases. In some instances, threads like PDO, despite having a smaller elastic limit, possess a larger elastic modulus, providing a robust elasticity due to material rigidity [6,12]. This stiffness provides adequate tissue support in the initial stages of the procedure and helps maintain lifted skin and tissues in place to a

certain extent, even when the face is in motion. Therefore, understanding the differences in the firmness and flexible elasticity exhibited by products due to the elastic modulus and elastic limit of each thread material enables the selection of the most suitable thread based on the patient's facial condition and procedural goals. Additionally, blending thread materials appropriately can lead to harmonious outcomes.

### 5. Thickness of Thread: Tensile Strength

The tensile strength of a thread refers to the maximum force or tension a thread can withstand without breaking when force is applied by pulling from both ends. When the materials and forms of threads are similar, the thread diameter is indeed the most crucial factor determining the tensile strength of each individual thread (Figure 2) [29].

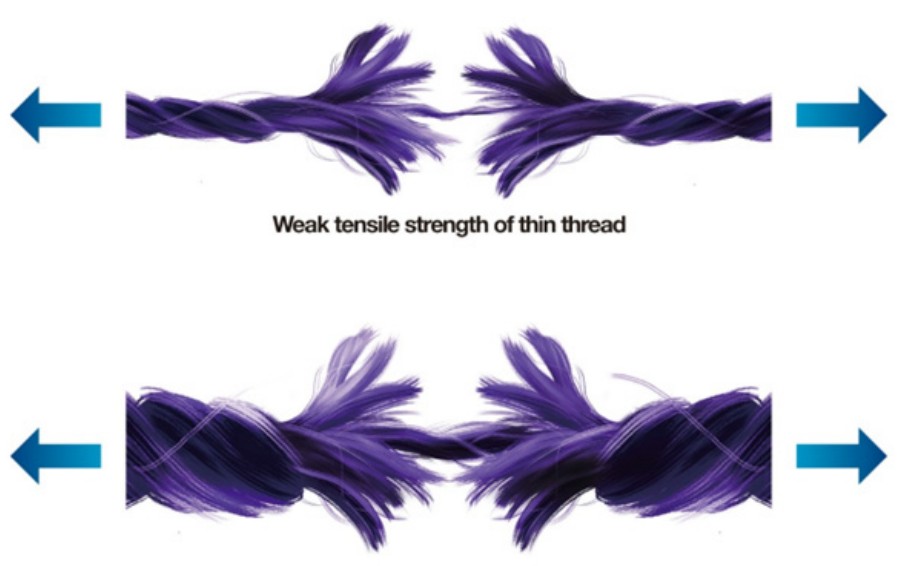

**Figure 2.** Relationship between thickness & and tensile strength of the threads. Tensile strength refers to the maximum force a thread can withstand without breaking when subjected to pulling forces from both ends. The arrows represent these pulling forces, highlighting how the thread's thickness, as indicated by its diameter, influences its ability to resist breaking under tension.

The threads used in thread-lifting procedures are typically manufactured in a cylindrical form. When comparing the size of elongated cylindrical objects, two methods are commonly used: comparing the diameter or circumference of the cylinder and comparing the cross-sectional area of the cylinder. Initially, the cross-sectional area of a cylinder can be determined using a mathematical formula: the area of a circle equals 3.14 multiplied by the square of the radius of the circle. After measuring the diameters of the cylinders and employing this formula to compute the cross-sectional areas of each cylinder, comparing the difference between the diameter and cross-sectional area reveals that the cross-sectional area of each cylinder varies proportionally to the square of the difference in diameters. For instance, if the difference in diameter between two cylinders is twice as large, their cross-sectional areas will differ by four-fold (2 squared equals 4), and if the diameter's difference is fourfold, their cross-sectional areas will differ by sixteen-fold (4 squared equals 16).

Upon measuring the tensile strength of threads with identical materials and shapes, comparing the differences in diameter and cross-sectional area of the threads reveals that the variation in tensile strength is proportionate not to the diameter but to the disparity in cross-sectional area. Typically, when categorizing threads based on thickness, one can observe the classification according to the United States Pharmacopeia (USP) size, which is primarily determined by the thread's diameter (Figure 3).

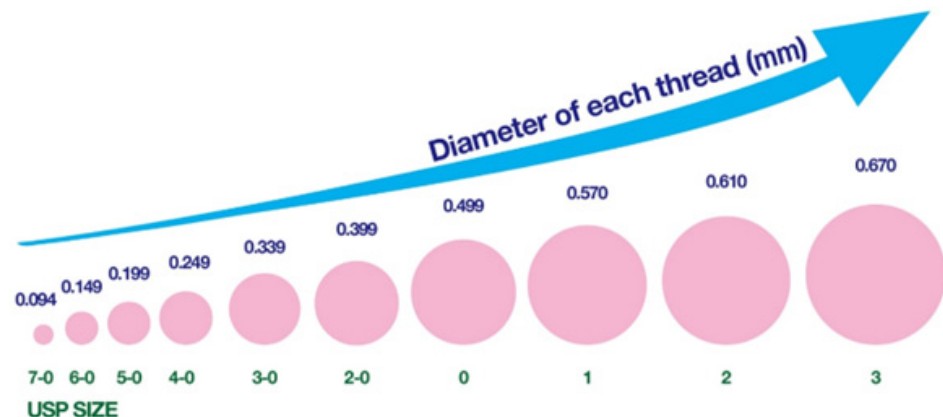

**Figure 3.** Comparison of thread thickness based on diameter. When examining threads made of the same materials and shapes, differences in tensile strength are not just proportional to the diameter but more so to the variations in cross-sectional area. Thread thickness classifications, often determined by United States Pharmacopeia (USP) size, are typically based on diameter variations.

According to this classification system, when comparing two threads with identical materials and shapes but differing thicknesses, if the diameter difference between the two threads is twice as large, the difference in their tensile strength is proportional to the square of the difference in their cross-sectional areas. For instance, suppose there are two threads made of the same material and shape, with one thread having a diameter of approximately 0.34 mm, classified as USP size 3-0, and the other thread having a diameter approximately twice as large, around 0.67 mm, categorized as USP size No. 3. Comparing their tensile strengths, on average, the 3-0 thread exhibits around 35 N, while the No. 3 thread demonstrates approximately 135 N of tensile strength. This comparison confirms that the larger diameter thread, which is double the size in diameter, experiences an approximately four-fold increase in tensile strength, which is proportional to the square of the difference in their diameters (Figure 4).

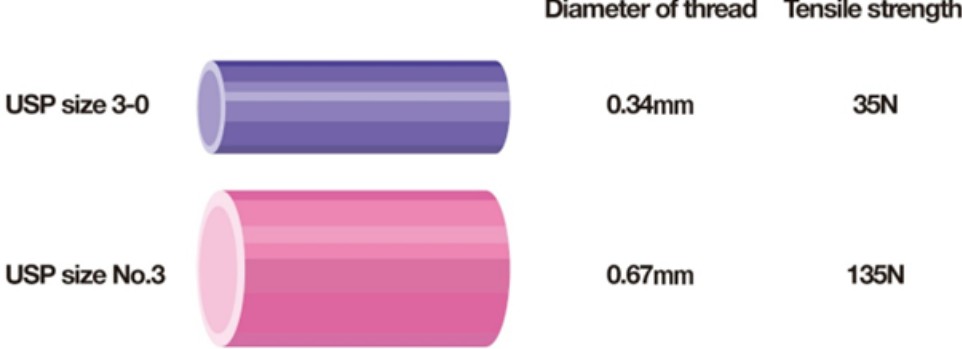

Cross-sectional area of thread = 3.14 x Radius of thread²
Tensile strength of thread - proportional to the cross-sectional area of thread

**Figure 4.** Differences in tensile strength according to differences in diameters of threads. For example, consider two threads of identical material and shape, where one measures around 0.34 mm in diameter, labeled as USP size 3-0, and the other thread is nearly twice the size, approximately 0.67 mm in diameter, designated as USP size No. 3. When assessing their tensile strengths, the 3-0 thread demonstrates an average of roughly 35 N, whereas the No. 3 thread showcases approximately 135 N of tensile strength. This comparison indicates that the larger diameter thread, which is double the diameter size, experiences an approximately four-fold increase in tensile strength, consistent with the square of the difference in their diameters.

The actual tensile strength of cogged threads used in thread lifting is determined not by the thread's original thickness but by the thinnest section of the thread after the formation of the cogs. Another crucial aspect to consider is the extent of heat treatment applied to the threads to create various shapes, including cogged threads. The heat introduced to the threads during this shaping process can alter the fundamental properties of the threads, resulting in a reduction in tensile strength. Additionally, even without deliberate heat application, mechanical manipulations to the threads to form cogs or specific shapes unavoidably generate heat, potentially impacting the material properties of the threads.

In comparing various cogged threads used in clinical settings, cutting-type cogged threads, created by machine cutting to form notches resembling thorns, result in sections where the thread's thickness becomes thinner due to these notches. Consequently, the overall tensile strength of the thread diminishes as a result of these sections rather than maintaining the original thread thickness (Figure 5).

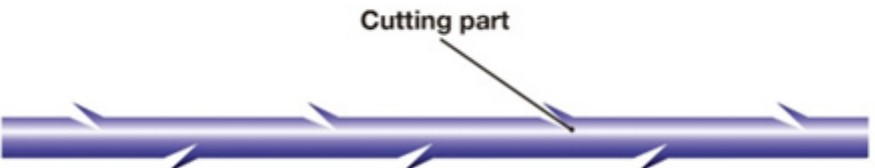

**Figure 5.** Cutting-type cogged thread. When contrasting different cogged threads employed in clinical practice, cutting-type cogged threads, crafted by machine cutting to shape notches resembling thorns, yield areas where the thread's thickness reduces due to these notches. As a consequence, the thread's overall tensile strength diminishes owing to these thinner sections rather than preserving the original thread thickness.

In recent times, there has been a growing trend in molding-type cogged threads aimed at preserving the thread's thickness [36,37]. However, it is important to note that these molding-type cogged threads may not always present exclusively advantageous features. Typically, such cogged threads are produced through two primary methods.

Firstly, there are molding-type cogged threads manufactured through an injection molding system, where the thread's material is placed into a molding mold of the cog shape, and heat is applied to attach the cogs. Throughout the manufacturing process of these cogged threads, the applied heat might cause slight distortions or deviations in shape (Figure 6).

Another method involves producing cogged threads through a press molding system. This method entails compressing a thicker thread material than the indicated size of the thread product using a sculpt cutter shaped like the cog pattern. Unnecessary parts are then trimmed and sculpted out from the thicker thread material, essentially stamping out the cogged pattern (Figure 7) [38,39].

It is essential to consider that the actual tensile strength of cogged threads used for lifting is determined not by the thread's thickness but by the thinnest part of the thread after creating the cogs. Another crucial consideration pertains to the degree of heat treatment applied to the threads, necessary to shape various thread products. During this process, heat application can alter the thread's fundamental properties, potentially reducing its tensile strength. Additionally, even without intentional heat treatment, heat generated during mechanical manipulation for shaping the threads might inadvertently impact the material's properties [2].

Upon comparative analysis of various cogged threads used clinically, those created using the cutting type, which involves carving the thread with machine blades, can result in reduced tensile strength due to surface indentations that create barbs, thinning specific areas of the thread and consequently reducing the overall tensile strength.

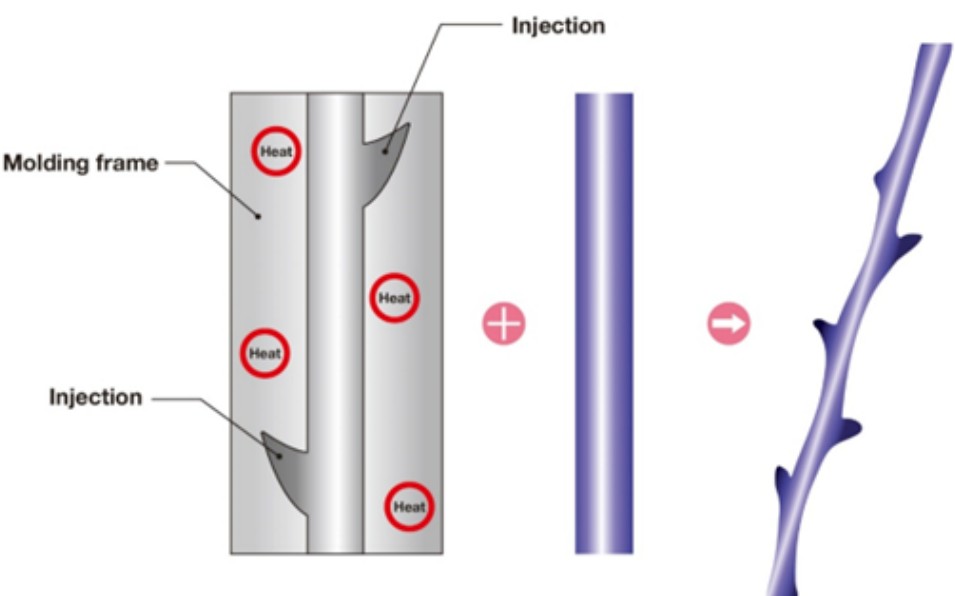

**Figure 6.** Manufacturing process of injection molding-type cogged thread. Initially, there exist molding-type cogged threads produced via an injection molding process, where the thread material is inserted into a cog-shaped mold, and heat is utilized to affix the cogs. In the manufacturing phase of these threads, the applied heat might induce minor shape distortions or discrepancies.

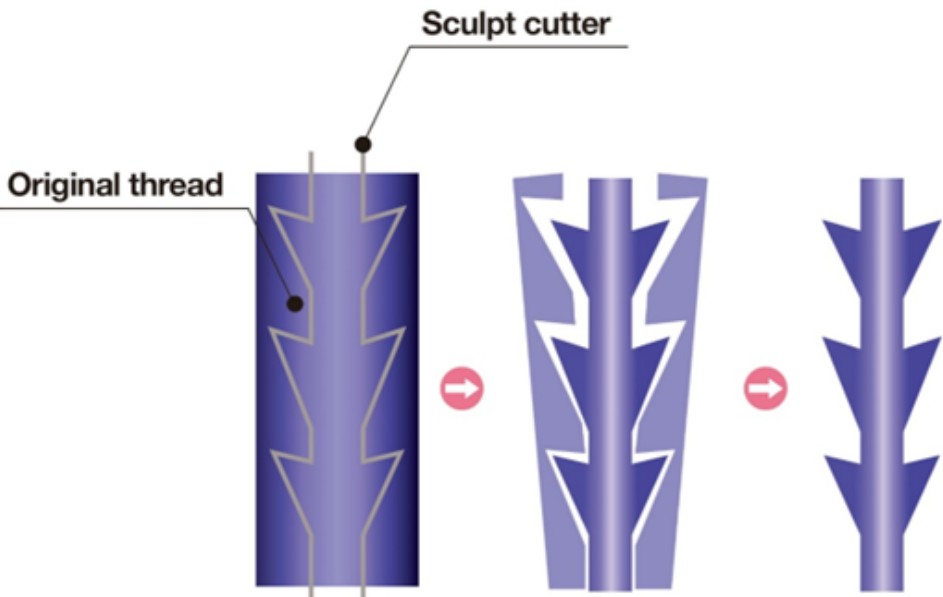

**Figure 7.** Manufacturing process of sculpt molding-type cogged thread. An alternative technique is the creation of cogged threads via a press molding system. This process involves compressing a thicker thread material using a sculpt cutter shaped like the cog pattern specified for the thread product. Subsequently, excess material is removed, shaping the cogged pattern through stamping.

Presently, there is a growing trend towards molding-type cogged threads to maintain the thread's thickness. However, not all molding-type cogged threads necessarily offer advantages. These threads are generally produced using two main methods: injection molding and press sculpt molding.

Injection molding-type threads, while having the same thread thickness as cutting-type threads, tend to exhibit higher tensile strength than those created with the cutting method due to the process of attaching cogs. However, those threads that undergo considerable

heat treatment during manufacturing might paradoxically show weaker tensile strength compared to cutting-type threads with minimal heat generation [40].

To prevent a decline in tensile strength due to heat treatment, many products known as heatless molding cogged threads have emerged, avoiding heat treatment during the production process. Nevertheless, it is essential to acknowledge that even with press molding techniques, some heat generation inevitably occurs during the cog shaping process, albeit in minimized amounts.

When utilizing molding-type cogged threads, one must consider the discomfort experienced during the insertion of the threads into the cannula and the subsequent procedure. Threads without cogs have sizes that are relatively similar, with slight differences in size due to the upward protrusion of the cut barbed cogs in cutting-type threads. In contrast, threads produced through injection molding have significantly larger sections with cogs compared to the product size. Sculpt molding-type threads, created by cutting out unwanted parts from thicker threads, result in portions without cogs being as thin as the product size, while the areas with cogs remain notably thicker, resembling the original thread (Figure 8).

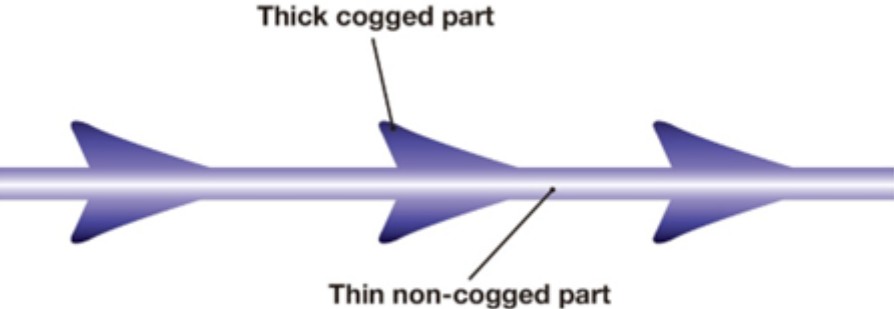

**Figure 8.** Sculpt molding-type cogged thread. Threads designed using sculpt molding involve removing sections from thicker threads, resulting in thinner segments lacking cogs, while areas with cogs maintain the original thread's thickness.

Hence, when performing procedures using cannulas, the cannula size required might increase with molding-type threads compared to cutting-type threads. This not only causes discomfort for the practitioner but also for the patients undergoing the procedure. Additionally, the thickness of the threads with cogs might cause a sensation of thickness greater than the product size, leading to patient discomfort during facial movements. Consequently, when utilizing molding-type cogged threads, careful consideration of the thread's size is imperative to address these concerns.

## 6. Shape, Location, and Number of Cogs: Anchoring and Holding Strength

The tensile strength of barbed threads is, in fact, a value that indicates how well a thread can withstand pulling forces equally from both ends under the same conditions without breaking, irrespective of several variables such as how securely the threads are lodged in the tissues or how sturdy the tissues where the threads are anchored are. As mentioned earlier, only the thickness of the thread and the extent of property changes due to heat generated during the manufacturing process can influence this value [41].

However, pulling force and maintaining force are characteristics unique to barbed threads as these forces manifest when the threads are actually engaging with the tissues. Pulling force, quite literally, refers to the force that, upon the thread's barbs coming into contact with the tissue, acts to secure and gather the tissue as if dropping an anchor in place [42]. On the other hand, maintaining force represents the value indicating how effectively these forces, once the threads are engaged with the tissues, persist without losing their effectiveness. Threads without barbs do not engage with tissues, resulting in the absence of pulling force and thus the absence of maintaining force that signifies the force the threads can withstand after engagement with tissues [12,35,41–43]

Some individuals may consider pulling force and maintaining force as a single entity or amalgamate them into just maintaining force. However, personally, the authors consider that differentiating these two forces is more clinically useful when selecting barbed threads. For instance, when contemplating the shape or size of the barbs, one may perceive that thinner and longer barbs have sharper edges and greater surface area in contact with tissues, making it easier for the barbs to engage with the tissues. Therefore, such barbs could be described as having better anchoring and anchoring strength compared to shorter barbs with blunt ends (Figure 9).

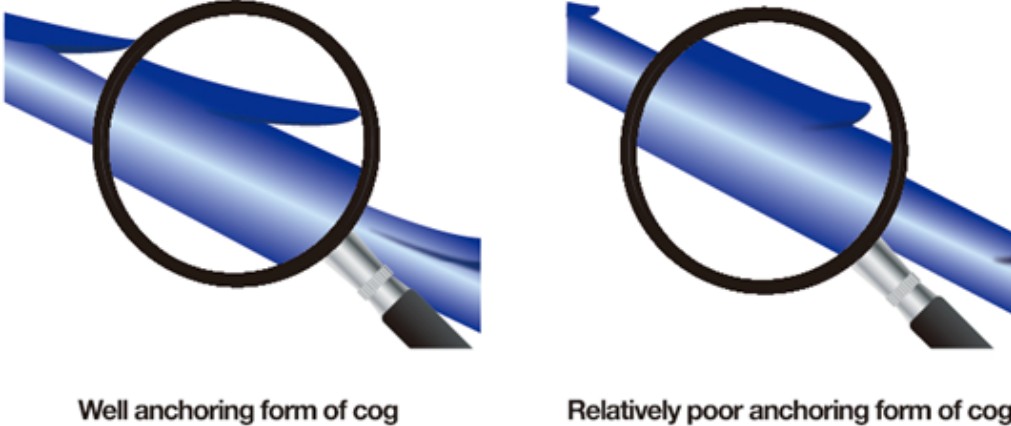

Well anchoring form of cog          Relatively poor anchoring form of cog

**Figure 9.** Distinguishing between pulling force and maintaining force is more practical when selecting barbed threads. Thinner, longer barbs with sharp edges and increased surface area offer better tissue engagement and anchoring strength compared to shorter, blunter barbs.

However, assuming that the barbs are adequately engaged with the tissue, the maintaining force, which signifies how effectively the force pulling and gathering the tissue is sustained, can be inversely influenced by the shape of the barbs. Thinner and longer barbs may exhibit decreased endurance compared to thicker barbs, hence the probability of a decrease in the force that holds and sustains the tissue over time when subjected to forces opposing the tissue traction due to facial movements and tissue loads is higher in such barbs (Figure 10).

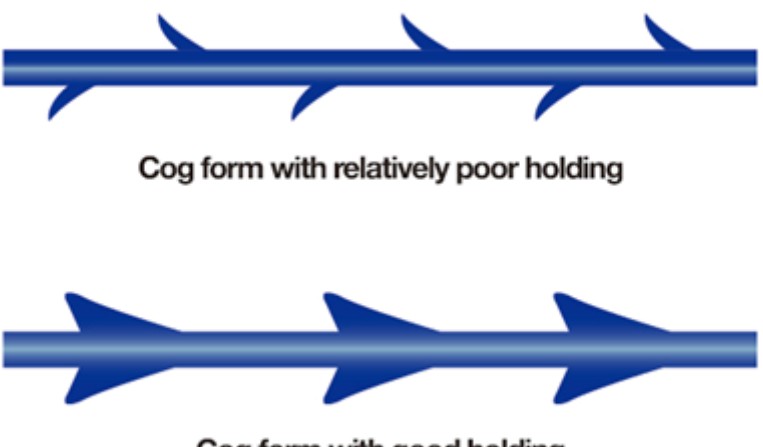

Cog form with relatively poor holding

Cog form with good holding

**Figure 10.** Thinner and longer barbs might have reduced durability compared to thicker ones. This can result in a higher likelihood of decreased force that supports tissues over time when facing opposing forces from facial movements and tissue loads.

Therefore, it is not desirable for the force to be biased toward one side. Ideally, during the initial insertion of the barbed threads, a strong sense of traction should be felt due to efficient anchoring of the threads to the tissue, facilitating immediate tissue grip and

pull. Subsequently, after the barbs have engaged with the tissue, a good barbed thread would be one that can sustain the pulling force while resisting the opposing forces without diminishing the traction exerted on the tissue. However, a thicker and sharper-ended barb is not necessarily superior. This is because excessively large or sharp barbs may be palpable externally, especially in patients with thin skin, and their thickness and length might necessitate larger cannulas during the procedure. Hence, when selecting barbed threads for a procedure, it is important to ensure that the chosen threads can effectively grip and sustain the expanded skin and tissue while possessing barbs of an appropriate size and shape for the procedure.

Another factor that can influence the traction and maintaining force of the barbs is their placement. Assuming all other conditions are identical for a single bidirectional cogged thread, it can be said that when the barbs are positioned on both sides of the thread as in a twin-sided type or in a spiral-sided type encircling the outer surface of the thread, the probability of tissues engaging with the barbs increases compared to a single-sided type where the barbs are only on one side of the thread. When barbs are positioned on multiple sides of the thread even after engaging with the tissue, the stress resisting the force opposing the traction provided by the barbs can be distributed across multiple parts of the thread, hence enhancing the maintaining force of the barbs (Figure 11) [44–47].

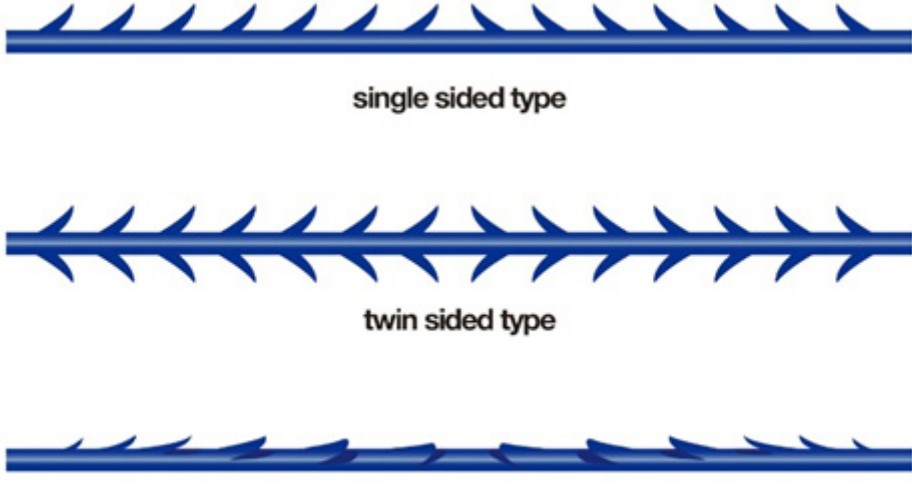

**Figure 11.** Barb positioning significantly impacts traction and holding force. Barbs placed on both sides or encircling the thread enhance tissue engagement, unlike a single-sided configuration. Multiple-sided barb placement distributes opposing stress, bolstering their maintaining force. Single-sided type (Secrete line AL cannula cog, Secrete line, Hyundai Meditech., Inc., Wonjusi, Republic of Korea), Twin-sided type (Secrete line Illusion, Hyundai Meditech., Inc., Wonjusi, Republic of Korea), and Spiral-sided type (Secrete line L cannula cog, Hyundai Meditech., Inc., Wonjusi, Republic of Korea).

Another crucial factor that influences the tensile and sustaining force of barbed threads under similar conditions is the number of barbs. When considering threads with equal overall length, a higher total number of barbs tends to increase the likelihood of effectively engaging with tissues and maintaining a strong pulling force even after engagement. The quantity of barbs becomes particularly significant when performing procedures on older patients with thin, weak tissues and pronounced sagging. In contrast, patients with relatively firm skin, retaining some elasticity and minimal sagging, might achieve surprisingly favorable results even without using densely packed or extensively positioned barbs. This is attributed to the comparatively resilient and sturdy nature of the skin and connective tissues, allowing the barbs to effectively engage with the tissues without

causing tissue tearing, thereby maintaining sufficient pulling force, even with a slightly lower number of barbs.

However, in cases where the skin tissues are soft, delicate, and less resilient, a higher number of barbs becomes essential. In such instances, a greater quantity of barbs is required to effectively engage with the tissues, ensuring that the pulling force is well-exerted. Moreover, after engagement, having numerous barbs distributed across multiple areas ensures that the pulling force remains consistent and does not weaken, enhancing the probability of maintaining adequate traction from various points and reducing the likelihood of force dissipation after pulling.

Therefore, for the same thread, long-length U or V-shaped bidirectional cogged threads, capable of performing the role of two threads due to their extended length, can exhibit stronger traction and sustaining force compared to short- or medium-length cogged threads, owing to the higher overall number of barbs, irrespective of the specific shape or size of individual barbs.

Hence, regardless of the chosen type of barbed threads when other conditions are similar, to achieve the effect of maximally and uniformly strengthening lax tissues, it is essential to either use barbed threads that are well-distributed across various sides of the thread or increase the number of inserted barbed threads. Ultimately, this aims to augment the total count of engaging barbs, ensuring the maximum firmness possible on the tissues being pulled.

## 7. Direction of Cogs

As previously described, the threads used in thread lifting are primarily employed in elongated forms based on their shape and length. They are utilized to gather tissue without a central cog by having the cogs on the left and right sides of the thread facing in one direction, known as the "long length bidirectional cogged thread" type. Additionally, threads are available in various designs based on cog orientation, predominantly adopting a linear form, classifying them as the "short or medium length cogged thread" type. While the direction of cogs in longer thread forms is generally predetermined, the discussion here focuses on the short- or medium-length cogged threads, which exhibit varied mechanisms based on cog orientation and consequent procedural objectives [14,47,48]

Previously, short- or medium-length cogged threads included unidirectional cogged threads where all cogs were oriented in one direction. However, these are seldom used for lifting purposes in contemporary practices. Instead, they are commonly categorized into the "single bidirectional cog" type, where opposing cogs face towards the central region, and the "multidirectional cog" type, where cogs face multiple locations, not necessarily centralized within the thread. Notably, an exception exists in the case of single bidirectional cogged threads used on the nose, where their orientation pulls the skin outward from the center rather than gathering it centrally, creating a longer appearance. This variant is termed the "reversed single bidirectional cogged thread" [49]. While multidirectional cog types can be further delineated based on their forms, such as double bidirectional, triple bidirectional, or multiple bidirectional types, for convenience, they are collectively referred to as multidirectional cog types owing to the orientation of cogs spanning more than a single location within the thread (Figure 12) [27].

When comparing the characteristics of these two types of threads, the short- or medium-length bidirectional cogged threads are designed with cogs oriented towards each other around the central empty part, aiming to gather tissue towards the center. This design ensures that the force exerted by the cogs is concentrated towards the center without dispersion, resulting in effective tissue suspension. However, a potential drawback of this design is the possibility of creating a bulging appearance when tissues are gathered towards the center.

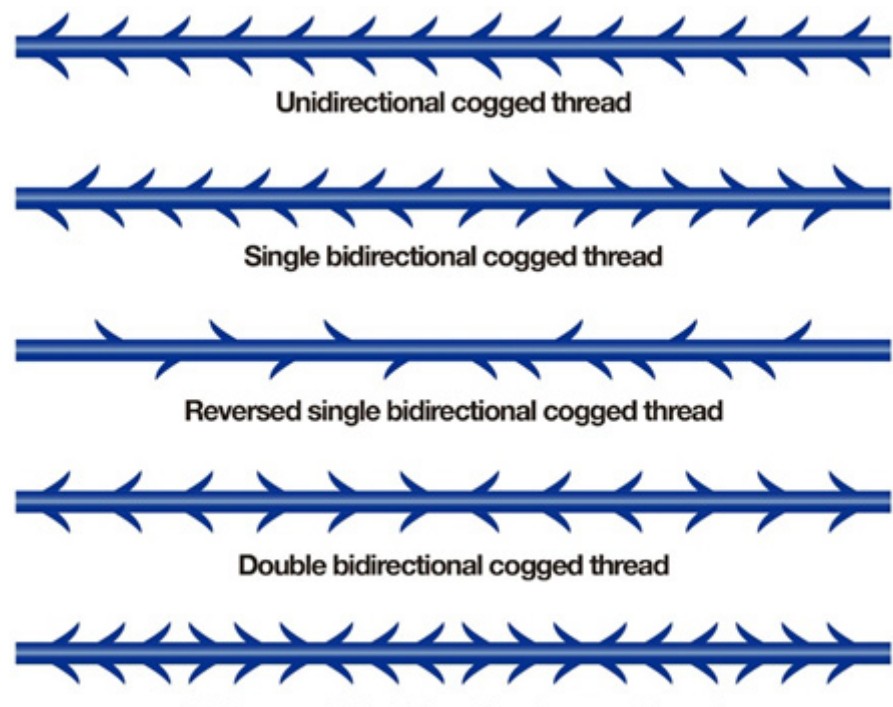

**Figure 12.** Previously, short- or medium-length cogged threads typically had unidirectional cogs, facing in one direction, but these are now rarely used for lifting. They are now commonly grouped into two main types: "single bidirectional cogs", with cogs opposing each other toward the center, and "multidirectional cogs", having cogs facing various locations along the thread. However, a reversed single bidirectional cogged thread is an exception, used on the nose, where its orientation pulls the skin outward, creating a lengthened appearance. Multidirectional cog types, including double, triple, or multiple bidirectional types, are collectively termed multidirectional cogs due to their cogs spanning more than a single location within the thread. The thread depicted in the image is the Secrete line Illusion (Hyundai Meditech., Inc., Wonjusi, Republic of Korea).

To utilize these characteristics advantageously in clinical settings, inserting I-shaped threads vertically from top to bottom in subzygomatic hollow patients can be beneficial (Figure 13). In this scenario, positioning the empty central portion of the short- or medium-length bidirectional cogged thread in the subzygomatic depression below the zygomatic arch is ideal. Aligning the entry point of the thread near the hairline, in line with the thread's length, allows for optimal placement in this area.

During the procedure, areas on the lower facial region that appear loose and bulging can be addressed by the threads, resulting in a tightening effect and flattening of these regions. Conversely, areas below the zygomatic arch experience tissue consolidation, improving depressed regions and creating an overall smoother appearance along the facial contour. However, a key consideration is the positioning of the thread's central part. If placed below the depressed area, it may cause the previously sunken area to appear more hollow or indented. Conversely, if the central part of the thread is positioned above the zygomatic arch, it might protrude, potentially affecting the prominence of the cheekbones. Thus, precise positioning is crucial to achieve the desired outcome.

However, as observed in the initial designs of short- or medium-length bidirectional cogged threads, when the number of protrusions on either side is equal around the center without protrusions, inserting the thread from top to bottom results in stronger force exertion on the upper protrusions engaging the firm tissues near the zygomatic arch, while the lower protrusions engage relatively loose cheek tissues, displaying weaker pulling force and sustaining power. Particularly in patients with relatively elongated faces, when positioning the midpoint gathering of protrusions just beneath the zygomatic arch to soften

the facial side contour, the shortened length of the thread placed on the lower cheek results in a relatively diminished number of engaging protrusions responsible for pulling the loose tissue in the lower area. Consequently, the force applied to the lower lax tissue weakens, as explained concerning the number of protrusions earlier. Given that a higher number of protrusions engaging the lower lax tissue is essential to effectively pull and gather the lax tissue, some companies are introducing improved versions of short- or medium-length bidirectional cogged threads. These threads are inserted from the top towards the face, featuring an unequal number of protrusions on both sides of the center, with an increased count of protrusions at the end of the thread placed in the lower lax area and a reduced count of protrusions on the handle side. This adjustment aims to enhance the direction of protrusions, ensuring sufficient tissue lifting effects on lax tissues. (Figure 14).

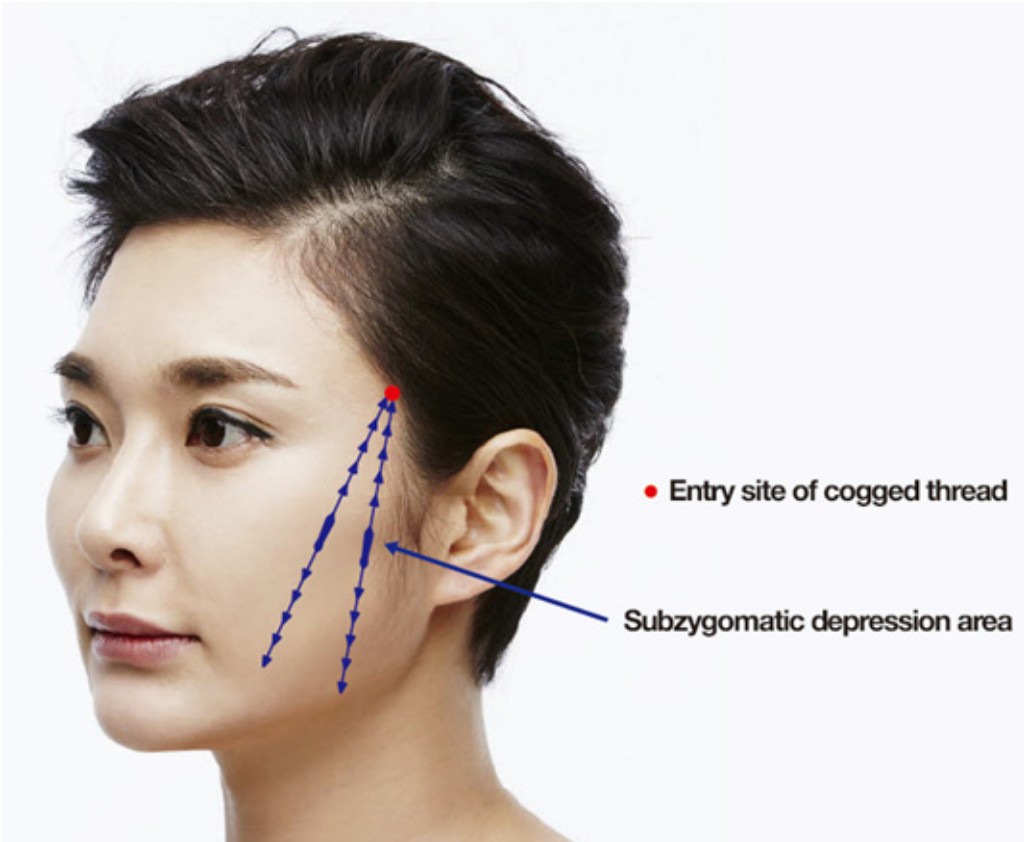

**Figure 13.** In clinical practice, inserting I-shaped threads vertically in subzygomatic hollow areas can be advantageous. Placing the central empty portion of bidirectional cogged threads under the zygomatic arch's depression, aligning the thread's entry point near the hairline, facilitates optimal placement in this region. The thread depicted in the image is the Secrete line Illusion (Hyundai Meditech., Inc., Wonjusi, Republic of Korea).

Compared to the typical bidirectional cogged thread that gathers tissues at only one central point, multidirectional cogged threads secure tissues by engaging protrusions across multiple areas. Therefore, it can be argued that the pulling force or lifting capacity, which pulls tissues strongly and gathers them solely at a central point, is diminished. However, in line with current trends, when patients are positioned lying down, with the chin lifted and the head facing downwards to induce natural movement of facial tissues due to posture change before inserting fixation threads, the ability of multidirectional cogged threads to uniformly secure tissues overall, preventing the previously moved tissues from shifting, is superior. Hence, it is more suitable for producing a general natural lifting effect in patients with less severe sagging. Presently, there is a growing trend in employing combination treatments that maximize the advantages of both types of threads,

initially utilizing bidirectional cogged threads to gather tissues and subsequently using multidirectional cogged threads to ensure the maintained position of the gathered tissues.

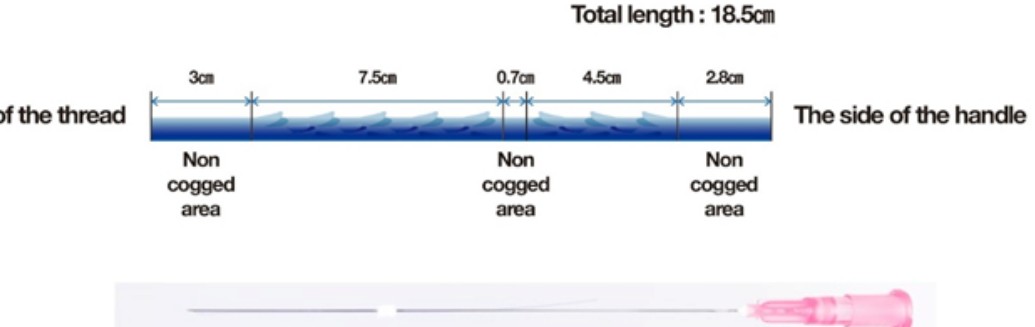

**Figure 14.** Therefore, to improve tissue gathering, companies are introducing enhanced bidirectional cogged threads with an uneven distribution of protrusions. This aims to maximize tissue lifting effects by increasing protrusions at the lower lax area and reducing them on the handle side during insertion from the top towards the face. The thread depicted in the image is the Secrete line Reverse (Hyundai Meditech., Inc., Wonjusi, Republic of Korea).

## 8. Insertion Depth of the Threads According to the Toughness of Tissues

When discussing the tensile and holding capabilities of cogged thread lifting, it was mentioned that one of the procedural conditions that can influence the outcome of cogged thread lifting is the type of tissue the thread engages with and how the patient's skin and tissues respond to it, thereby affecting the degree of tensile strength and holding power exhibited by the thread. For instance, even when the same cogged thread is used, if the thread engages with resilient tissues such as the superficial musculoaponeurotic system (SMAS), it shows greater tensile strength and holding power compared to instances where the thread enters the subcutaneous fat layer with little tissue resistance (Figure 15).

Determining the layer of tissue where the thread enters during cogged thread lifting is crucial not only for showcasing lifting effects but also for minimizing the side effects of the thread. Typically, areas around the head and ears are composed of firm tissues, necessitating the insertion of threads into deeper tissues like SMAS to prevent threads engaging the lower loose tissues from sagging again. However, for the lower threads that cause sagging, the threads should be inserted to engage with tissues comprising the skin and subcutaneous fat layers that contribute to sagging. Even within the same subcutaneous fat layer, areas closer to the skin exhibit a greater presence of fibrous tissues formed by the proliferation of ligamentous tissues. Hence, when the thread is closer to the skin, it adheres better and retains a firmer grip even after engagement. Conversely, if the thread only enters the subcutaneous fat very close to the skin, the pulling force of the thread does not transmit to the deep subcutaneous fat, affecting only the subcutaneous fat connected to the outer skin, leading to a higher likelihood of the skin surface resembling a dimple or being pushed altogether due to the skin bitten by the thread. Furthermore, when the skin is thinner than the thread's thickness, the outline of the thread may appear on the skin surface.

Generally, when using cogged threads thicker than 1-0, if the threads enter only the subcutaneous fat close to the skin, the skin can appear pushed out, increasing the potential for dimpling. Hence, it is most ideal to insert the threads into SMAS tissue or the subcutaneous fat layer close to the SMAS. Although fibrous septa are also present in the subcutaneous fat layer located in the middle, threads engaging around the SMAS exhibit weaker gripping forces compared to threads engaging the SMAS itself. Therefore, when using cogged threads with a thickness of 1-0 or more, efforts should be made to insert the threads around the SMAS or close to it to reduce the side effects of shallow insertion and maximize the lifting effect of tissue pulling. In practical terms, during cadaver studies for thread lifting confirming the insertion of cogged threads around the SMAS, through or

surrounding the SMAS was deemed most suitable after insertion at what was considered the most appropriate location (Figure 16).

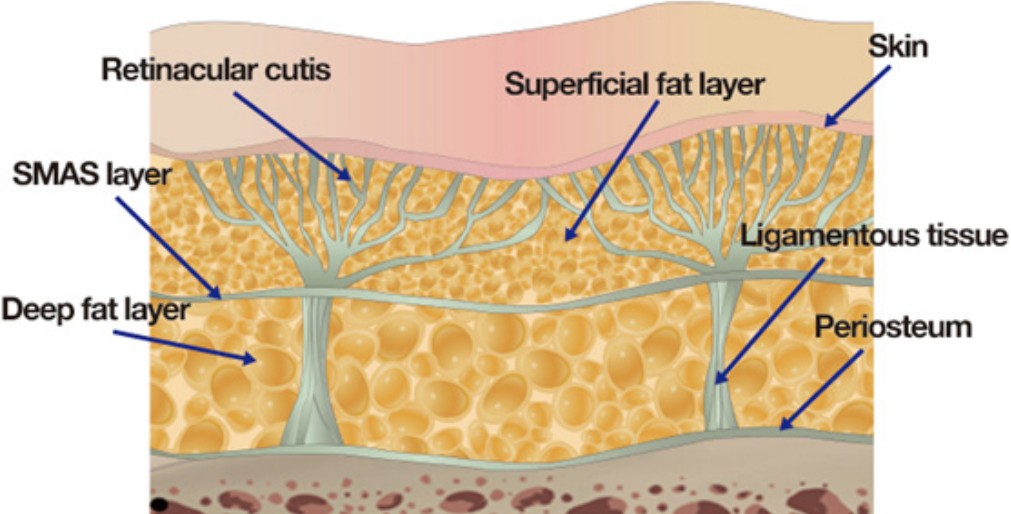

**Figure 15.** The outcome of cogged thread lifting depends on tissue engagement, patient skin response, and the resulting tensile strength. For instance, engaging resilient tissues like SMAS yields greater thread strength than less-resistant fat layers.

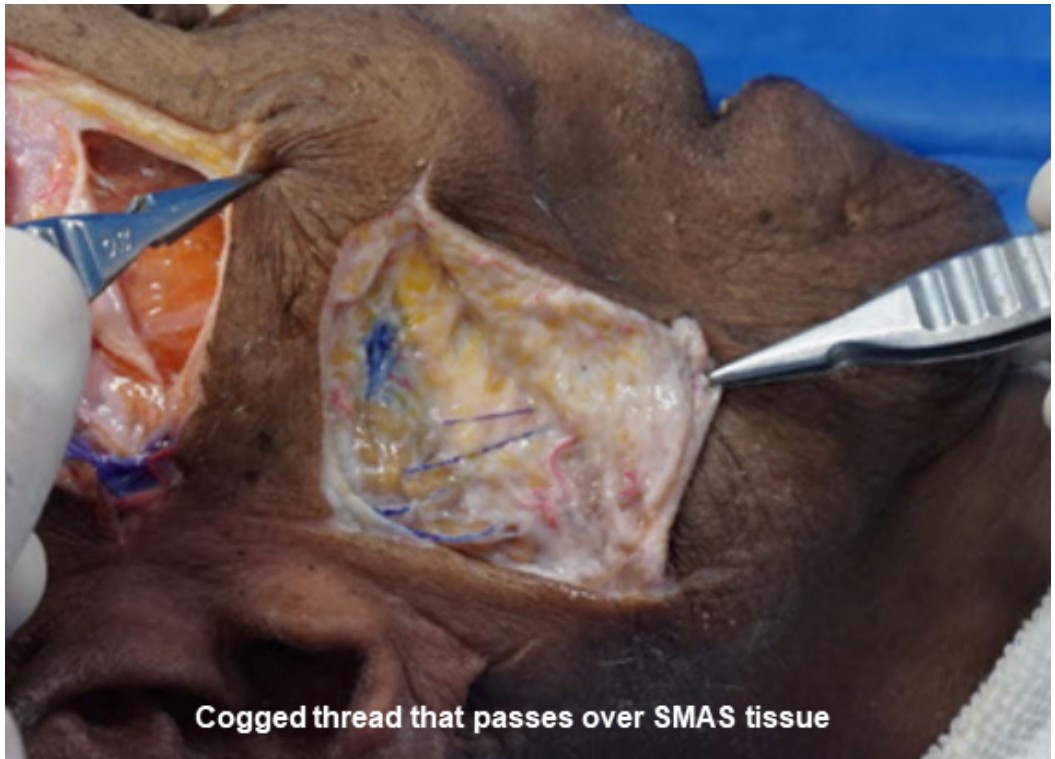

**Figure 16.** Hence, when utilizing cogged threads of 1-0 thickness or greater, it is advisable to place these threads near or around the SMAS (superficial musculoaponeurotic system) to minimize potential side effects of shallow insertion and enhance the tissue-pulling lifting effect. In practical studies involving cadavers for thread lifting, it was found that confirming the insertion of cogged threads around or adjacent to the SMAS post-insertion was the most optimal placement strategy. The thread depicted in the image is the Secrete line Multi-cog (Hyundai Meditech., Inc., Wonjusi, Republic of Korea).

### 9. Selection of Appropriate Lifting Vector and Fixing Point according to Facial Area

As facial tissues succumb to aging and sagging, it is observed that not all tissues sag uniformly. Tissues surrounding the head and ears, which are firmly adhered, tend not to sag as much, whereas tissues around the cheeks, mouth, and outer jawline, which lack firm internal structures, tend to sag more readily (Figure 17) [22].

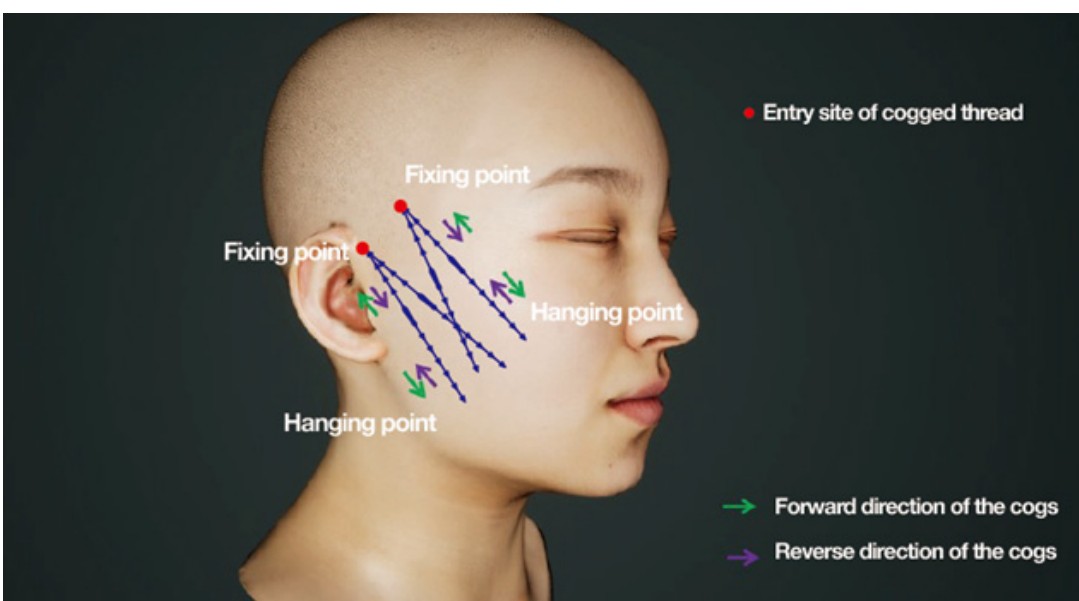

**Figure 17.** To effectively employ these points for securing threads in sagging tissues, it is crucial to adjust the direction of the threads based on the specific tissues requiring lifting and repositioning. This entails deciding the appropriate orientation of the threads towards the fixation point, depending on the particular tissue requiring relocation and repositioning. The thread depicted in the image is the Secrete line Illusion (Hyundai Meditech., Inc., Wonjusi, Republic of Korea).

Therefore, adhering to the fundamental principles of thread lifting while maximizing effectiveness and longevity necessitates moving away from the notion of merely pulling the underlying tissues upward and, instead, strategically relocating lax tissues that tend to sag due to aging towards firmer tissues that resist sagging structurally. Efforts should be made to ensure that once repositioned, these tissues are well adhered to the new location to maintain their position.

Hence, utilizing threads to effectively lift sagging and lax tissues is not the end goal. It is crucial to identify fixing points within relatively immobile and firm tissues to prevent threads from loosening again. In the temporal and periauricular regions, there exist robust tissues such as the temporal fascia, temporal septum, zygomatic ligament, Lore's fascia, and platysma auricular ligament, which can serve as suitable fixing points [50–53]. Proficient anatomical understanding and utilization of these firm tissues can enhance the effectiveness and duration of thread lifting [22].

To optimally utilize these thread-fixing points for the sagging tissues, it is essential to consider varying the vector to which the threads are directed based on the target tissue that needs to be lifted and repositioned. This consideration aims to determine how the threads should be directed towards the fixing point differently depending on the target tissue that requires relocation and repositioning. Face lifting surgeries initially described pulling the skin and tissues of the mid and lower face upward and forward at an oblique angle. Over time, the concept evolved to advocate for vertical lifting rather than the oblique method, suggesting that a more effective lifting effect is achieved when tissues are pulled vertically. This concept has also been applied in thread-lifting procedures. Previously, the preferred direction of thread lifting was a slightly oblique design from bottom to top. However, the current preference leans towards vertical lifting designs [15,22].

It is important to note that the mere act of pulling tissues vertically in any area due to aging and gravity might not align with anatomical aging phenomena. While it is true that tissues tend to sag downwards due to gravity, considering the orientation and role of robust ligamentous structures that hold the skin and tissues in place alters the appearance of surrounding tissues and resulting wrinkles. Inserting threads vertically into these wrinkles might maximize the lifting effect.

For instance, the connective tissues on the lateral face tend to sag almost vertically toward the outer jawline. Thus, a vertical lifting approach, closer to vertical orientation, utilizing the robust tissues present in the temporal region, effectively pulls and secures sagging tissues. However, sagging tissues above the jowls and jowling wrinkles tend to develop wrinkles oriented towards the ears, suggesting that oblique lifting, pulling the sagging tissues towards the firm tissues around the ears using threads, might be more effective.

Furthermore, the firm mandibular ligament holding tissues can be effectively addressed by pulling the sagging tissues outward towards the zygomatic arch area near the ear. Considering these anatomical aspects and lifting tissues vertically, minimizing interference with facial structures, can be considered the most ideal lifting vector (Figure 18).

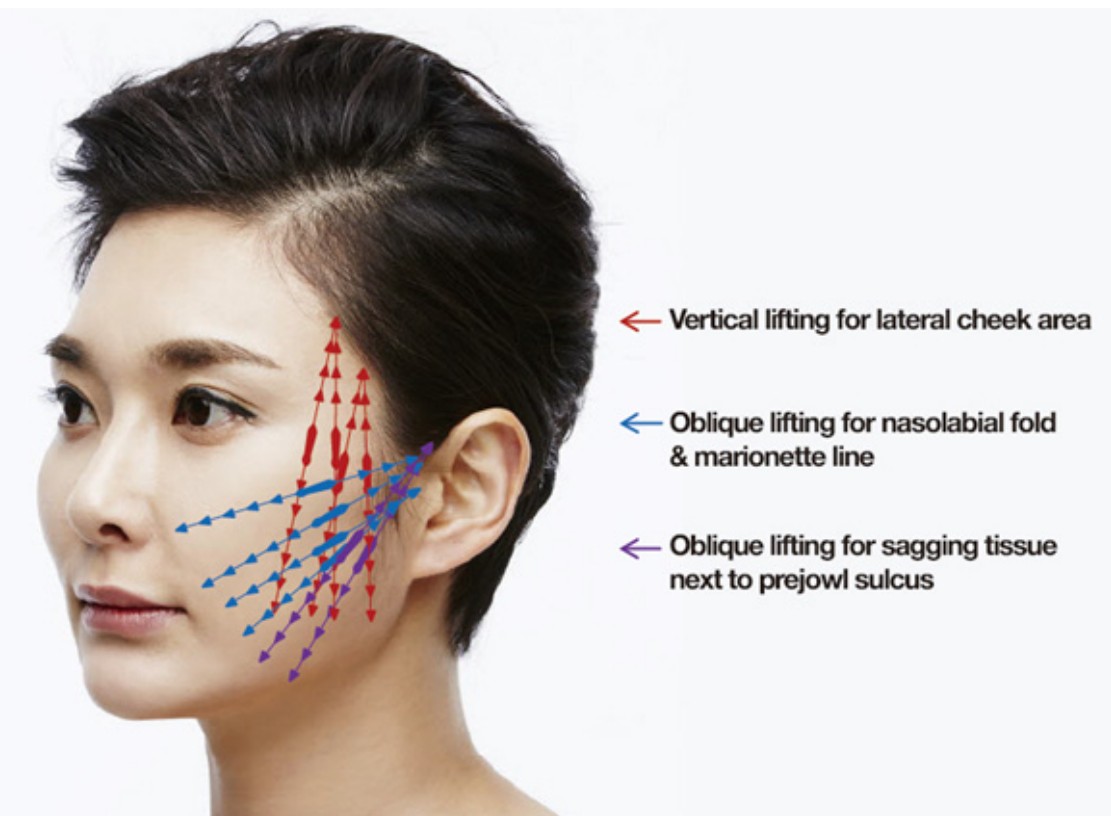

**Figure 18.** Moreover, addressing the robust mandibular ligament that secures tissues can involve pulling the drooping tissues outward towards the zygomatic arch region close to the ear. This lifting vector, mindful of these anatomical features and vertically elevating tissues to reduce disruption to facial structures, could be deemed the most optimal approach. The thread depicted in the image is the Secrete line Illusion (Hyundai Meditech., Inc., Wonjusi, Republic of Korea).

## 10. Difference between Flap and String Methods

When explaining the lifting procedure using cog threads, nowadays, the lifting effects can be broadly classified into two approaches based on the length of the threads, the shape of the cog elements, and their intended purpose.

The first approach resembles the shape of a skin flap used in skin reconstruction surgery, where skin tissue is pulled in one direction, akin to the single bidirectional cogs found in elongated U- or V-shaped threads. These threads function as a representative method of lifting by pulling skin tissue in a flap-like manner.

The second approach, unlike the flap technique, involves the cogs of the threads not pulling a fixed area of skin and tissue in one direction, but rather resembles a string technique. This method involves marking a portion of the sagging skin and tissue and then pulling it in the targeted direction by threading the strings. While employing this technique in areas with broad sagging skin and tissue may increase the number of points where the strings are applied, its advantage lies in the diverse design possibilities for each direction of pulling based on the facial structure or the pattern of sagging tissue (Figure 19) [54]. The evaluation of whether the flap method is better than the string method or vice versa cannot be deemed absolute, as it depends on the patient's condition and the purpose of the procedure. It is plausible to judiciously combine these two methods as per the requirement.

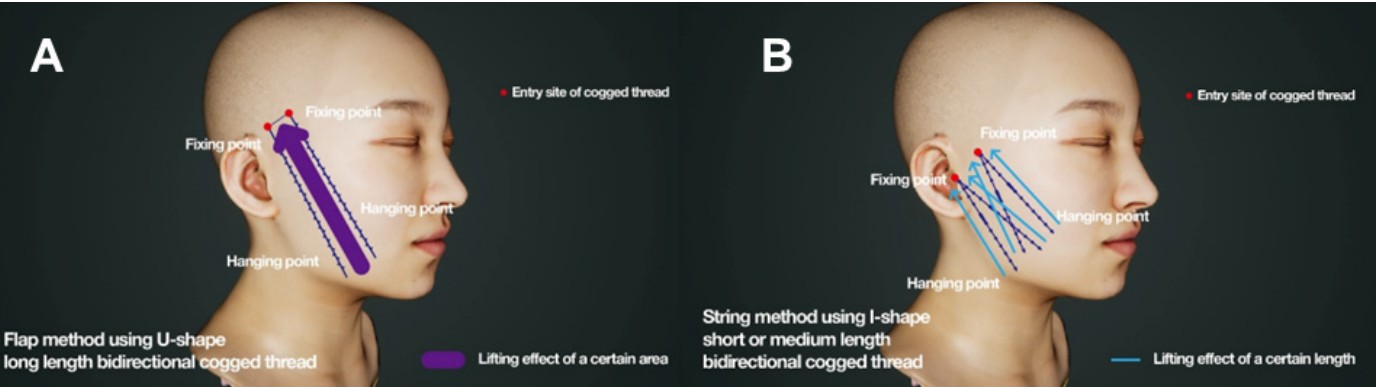

**Figure 19.** Method one utilizes threads with unidirectional cogs, resembling a skin flap, to pull tissue in one direction (**A**). Method two, unlike the skin flap approach, uses threads resembling a string technique to mark and pull sagging tissue in diverse directions (**B**). The thread depicted in the image is the Secrete line Illusion (Hyundai Meditech., Inc., Wonjusi, Republic of Korea).

From the author's personal standpoint, preference leans toward the string technique using shorter or medium-length threads, pulling the tissue in the opposite direction of sagging, as opposed to maximizing the pulling force with elongated threads in the flap method. When employing the string method, the choice between using bidirectional cogged threads, emphasizing tissue gathering force, or multidirectional cogged threads, focusing on maintaining the repositioned tissues according to the movement, can be made based on the condition of the patient's skin and tissue and the procedural objectives. Additionally, there might be situations where both types of threads are appropriately mixed for use.

When describing the lifting procedure utilizing cog threads, there is a prevalent division into two main methodologies based on the length of the threads, cog shape, and intended purpose that yield lifting effects.

The fundamental principle of flap-based thread lifting, whether using elongated U- or V-shaped threads or shorter I-shaped threads, revolves around configuring the cog's reverse direction towards the center part of the thread. This configuration aims to pull the skin and tissue outside the forward-facing cogs, using a pulling force, known as tensile strength, directed towards the center of the reverse side of the thread. This method prioritizes the tensile strength, emphasizing the pulling force generated to gather skin and tissue widely.

Employing this technique may initially impart a strong sensation of firmly holding the skin, resembling an intensive lifting procedure. However, this method is prone to a drawback where, due to the cogs' gathering towards the center, the maximum tension tends to accumulate at the thread's central area. Consequently, if the holding force weakens in this region, there is a rapid decline in the pulling effect on the tissues. Uneven distribution

of force among the cogs or a reduction in force on one side can disrupt the balance of force, leading to a sudden decrease in the pulling force gripping the skin and connective tissues.

An individual can sense that when sitting then slightly tilting the chin upward after reclining, the loose tissues around the cheeks and jawline tend to shift naturally toward the firm areas around the ears and the head [55]. The repositioning technique developed for natural thread lifting aims not to forcibly push the tissues beneath upward but rather redirects the loose tissues toward the firm areas, ensuring they maintain their position when returning to a sitting posture from a reclined one. The goal is to prevent these tissues from returning to their original position when the patient moves from lying down to sitting. While this method appears theoretically desirable, the challenge lies in ensuring the cog threads gripping the tissues do not release under the returning tissue load when the patient sits up. However, in the early stages of developing this method, the threads struggled to generate sufficient tension to resist and withstand the load exerted by the tissues seeking to return to their original position. Moreover, these initial threads lacked the strong pulling effect in one direction, resulting in a relatively decreased lifting effect on the lower and midfacial regions compared to the flap method. Consequently, they were limited to partial usage. Yet, with the enhancement in thread quality and cog manufacturing technology, threads adopting this repositioning method are increasingly being produced. Recent advancements have introduced string-style cog threads that maximize cog performance by adjusting cog shape, direction, and positioning.

## 11. The Expiration Date of Absorbable Threads

When performing dissolvable cog thread lifting, a crucial consideration pertains to the duration it takes for the most commonly used material, PDO thread, to completely dissolve within the human body. However, the actual fate of the thread after insertion into the body remains speculative, varying across different time intervals. Typically, even as the thread begins to dissolve and soften, collagen stimulation occurs, contributing to skin and tissue regeneration. Hence, it is commonly understood that the skin tightening effects persist to some extent even after the initiation of thread dissolution. Nevertheless, there is a need for systematic research to ascertain whether the effects derived from collagen generation persist to a certain degree after the thread weakens and loses its pulling effect. Some medical professionals assert that once the thread weakens, despite collagen generation, the actual effects might exist only histologically, while clinically, they largely diminish. Further comprehensive studies are warranted to validate these claims.

In the process of manufacturing cog thread products, PDO threads are initially processed to create cogs before being inserted into needles or undergoing shape alterations. Following these steps, the final product is subjected to sterilization and sealed packaging. Typically, upon opening the packaging and inserting the PDO thread into the body through procedures, the thread begins immediate decomposition. Within a month or two, the thread tends to soften significantly, resembling cooked pasta, and around 6-8 months post-insertion, it completely dissolves and disappears. Upon opening the sealed packaging, the thread is exposed to moisture in the air, initiating an inconspicuous but gradual decomposition process. Therefore, when using PDO thread products left unused after opening the packaging for an extended period, the tensile strength of the threads significantly diminishes, resulting in threads breaking more easily. Even products left unused for only one or two days post-opening exhibit faster decomposition upon insertion into the body compared to those that remain unopened. Hence, due to the likelihood of exposure to moisture during inadequate manufacturing processes, thread products with suboptimal quality are prone to reduced immediate effects and a shorter duration of effectiveness following insertion into the body.

## 12. Discussion

The evolution of thread-lifting procedures has expanded beyond addressing aging-related skin and tissue laxity to accommodate a broader demographic, including younger

patients seeking refined facial contours. The introduction of cog threads has revolutionized these procedures, enabling sculpting effects beyond traditional lifting, impacting the midface, perioral areas, and jawlines. However, selecting the most suitable threads poses challenges due to varying patient needs and physician preferences [56–59].

While manufacturers often showcase dramatic effects, establishing clear indications for product efficacy could enhance credibility. Tailoring thread selection to individual cases based on tissue laxity remains crucial. Lighter cog threads suit minimal laxity, while stronger threads with robust traction are necessary for significant laxity, although no single thread type universally fits all cases.

Current trends favor combining various threads for optimal outcomes rather than relying on a single type. Nonetheless, excessive traction during procedures can lead to prolonged discomfort or skin irregularities. Mitigating these side effects requires a delicate balance between applied tension and post-procedural measures, acknowledging each thread type's advantages and limitations.

Therefore, an in-depth understanding of thread characteristics and their impact on outcomes remains pivotal. This understanding guides practitioners in selecting threads aligned with patient needs and procedural goals, transcending specific company products. By examining various factors influencing thread-lifting procedures, practitioners can strive for optimal results while maintaining the minimally invasive nature of these procedures. These pertinent factors are consolidated and presented in Table 1 for convenient reference and analysis.

**Table 1.** Factors influencing the outcome of thread lifting.

| **Factors Influencing the Outcome of Thread Lifting** |
| --- |
| 1. Materials of Threads |
| 2. Thickness of Thread: Tensile Strength |
| 3. Shape, Location, and Number of Cogs: Anchoring and Holding Strength |
| 4. Direction of Cogs |
| 5. Insertion Depth of the Threads According to the Toughness of Tissues |
| 6. Selection of Appropriate Lifting Vector and Fixing Point According to Facial Area |
| 7. Difference Between Flap and String Methods |
| 8. Expiration Date of Absorbable Threads |

**Author Contributions:** Conceptualization, G.-W.H., K.-H.Y. and S.-Y.P. Writing—Original Draft Preparation, G.-W.H., K.-H.Y., S.-Y.P. and J.W. Writing—Review and Editing, G.-W.H., K.-H.Y., S.-Y.P. and J.W. Visualization, G.-W.H., H.H. and K.-H.Y. Supervision, G.-W.H. and K.-H.Y. All authors have read and agreed to the published version of the manuscript.

**Funding:** This research received no external funding.

**Acknowledgments:** This study was conducted in compliance with the principles set forth in the Declaration of Helsinki.

**Conflicts of Interest:** I acknowledge that I have considered the conflict of interest statement included in the "Author Guidelines". I hereby certify that, to the best of my knowledge, that no aspect of my current personal or professional situation might reasonably be expected to significantly affect my views on the subject I am presenting.

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
