# Peer review of "What Are the Factors That Enable Thread Lifting to Last Longer?"

_cosmetics, doi:10.3390/cosmetics11020042_

Round 1
Reviewer 1 Report
Comments and Suggestions for Authors
What is the main question addressed by the research? I I think this study could be useful for those dealing with aesthetic dermatology
Is it relevant and interesting? yes How original is the topic? The topic is original What does it add to the subject area compared with other published material? Is the paper well written? the paper is well written Is the text clear and easy to read? the text is easy to read Are the conclusions consistent with the evidence and argumentspresented? the conclusion is consistent.
A fascinating paper on aesthetic dermatology, I have just a concern, I think it is useful to add a paragraph on possible side effects
Comments on the Quality of English LanguageGood quality
Author Response
We extend our gratitude for the time and effort all the reviewers devoted to providing feedback on our manuscript. The insightful comments and valuable improvements suggested have been acknowledged with appreciation. We have incorporated the reviewers' recommendations, and these enhancements are highlighted within the manuscript.
1. Kindly refer to the newly added Section 2, which comprises a paragraph discussing the side effects associated with thread-lifting.
Reviewer 2 Report
Comments and Suggestions for Authors
I congratulate you on a very good analysis. As a doctor, I am unfortunately missing an analysis of individual factors, diseases, skin conditions as also influencing the length of the lifting procedure effect. Have you at least analysed literature data on which skin and health conditions of patients can be expected to have a longer or shorter effect? I think this is essential. Please add such an element and the necessary references.
Author Response
We extend our gratitude for the time and effort all the reviewers devoted to providing feedback on our manuscript. The insightful comments and valuable improvements suggested have been acknowledged with appreciation. We have incorporated the reviewers' recommendations, and these enhancements are highlighted within the manuscript.
2. Section 3 addresses the key factors we deem essential in patient selection to ensure optimal and enduring outcomes in thread-lifting.